# Clinical Protocols and Treatment Guidelines for the Management of Maternal and Congenital Syphilis in Brazil and Portugal: Analysis and Comparisons: A Narrative Review

**DOI:** 10.3390/ijerph191710513

**Published:** 2022-08-24

**Authors:** Talita Katiane de Brito Pinto, Aliete Cristina Gomes Dias Pedrosa da Cunha-Oliveira, Ana Isabela Lopes Sales-Moioli, Jane Francinete Dantas, Rosângela Maria Morais da Costa, José Paulo Silva Moura, Sagrario Gómez-Cantarino, Ricardo Alexsandro de Medeiros Valentim

**Affiliations:** 1Laboratory of Technological Innovation in Health, Federal University of Rio Grande do Norte, Natal 59010-090, Brazil; 2Health Sciences Research Unit: Nursing (UICISA: E), Nursing School of Coimbra (ESEnfC), 3001-901 Coimbra, Portugal; 3Center for Interdisciplinary Studies of the 20th Century (CEIS-20), University of Coimbra, 3000-186 Coimbra, Portugal; 4Municipal Health Department, Natal City Hall, Natal 59014-030, Brazil; 5Doctor Daniel de Matos Maternity, Coimbra Hospital and University Center, 3000-157 Coimbra, Portugal; 6Faculty of Medicine, University of Coimbra, 3000-370 Coimbra, Portugal; 7Faculty of Physiotherapy and Nursing, Toledo Campus, University of Castilla-La Mancha, 45071 Toledo, Spain; 8Department of Biomedical Engineering, Federal University of Rio Grande do Norte, Natal 59078-970, Brazil

**Keywords:** syphilis, congenital syphilis, clinical protocols

## Abstract

(1) Background: Maternal syphilis (MS) and congenital syphilis (CS) are serious public health problems worldwide due to their high morbidity and mortality rates. (2) Objective: Evaluating the applicability of Clinical Protocols and Treatment Guidelines on case incidence trends in Brazil and Portugal. (3) Methods: The review was done through bibliographic research in two public databases and government websites from both countries, published between 2007 and 2022. All guidelines that contained CS and MS were selected. (4) Results and discussion: After evaluation, we found that Brazil and Portugal have adequate protocols for screening and treating congenital and maternal syphilis. (5) Conclusion: The results suggest that CS and MS incidence are notably higher in Brazil than in Portugal due to economic, cultural, and social disparities and the differences in territory size. Therefore, these demographic and socioeconomic factors could strongly influence efforts to fight against syphilis and thus control the infection.

## 1. Introduction

Syphilis is a systemic infection caused by *Treponema pallidum* subspecies *pallidum* (*T. pallidum*), a human pathogenic bacterium [1,2,3,4,5]. The World Health Organization (WHO) estimates nearly 6 million new infections globally each year, with prevalence highest among people aged 15–49 years [6,7]. If not detected early, the infection can progress to its later stages and too many severe clinical manifestations, including neurosyphilis, blindness and cardiovascular disorders [3,8,9]. Syphilis is primarily spread through unprotected sexual contact, which can also lead to other sexually transmitted infections (STIs), such as human immunodeficiency virus (HIV) [7,10,11]. In addition, congenital transmission can occur when an untreated or poorly treated pregnant woman with syphilis transmits *T. pallidum* onto her child through the placenta (approximately 80% of cases)—which can occur at any time in pregnancy or any disease stage—or occasionally through direct contact with syphilitic sores when the baby moves into the birth canal [12,13,14].

According to WHO [15], congenital syphilis is the second-leading cause of stillbirths worldwide, surpassed only by malaria, and can pose significant health problems, including miscarriage, preterm births, and congenital infection in newborns, as the disease leads to severe sequelae if left untreated [3,9,16,17,18,19,20]. According to a 2017 estimate by WHO [17], nearly 930,000 pregnant women get infected with syphilis worldwide yearly, resulting in approximately 143,000 early fetal deaths/stillbirths. Notably, most CS cases are asymptomatic and concentrated in the early stages of infection (about 70%); however, infected newborns may have multiple clinical manifestations. Some manifestations include low birth weight, hepatomegaly or splenomegaly, osteochondritis, maculopapular skin lesions, pseudoparalysis, generalized lymphadenopathy, seizure, leukopenia, or leukocytosis.

The standard treatment recommended by the WHO is Benzathine Penicillin G (BPG), strongly indicated to treat pregnant women with syphilis during prenatal care, and crystalline or aqueous penicillin, a drug indicated for the treatment of children born with congenital syphilis. These are low-cost and easily accessible medicines available in Basic Health Units. However, despite the ease, there was a shortage of these drugs between 2014 and 2016, which affected several countries in the world, such as Australia, Canada, Croatia, Germany, Greece, Netherlands, Switzerland, United States, and Brazil (60.7% of Brazilian states) [21,22,23].

This lack of medication compromised the treatment of confirmed cases of pregnant women with syphilis, which was reflected in the increase in congenital syphilis cases in Brazil during this period [24,25]. In 2015, the Pan American Health Organization (PAHO) declared that 22 countries in the Americas reported treatment of positive pregnant women, and, in the following year, the lack of BPG was identified as one of the causes of the increase in CS cases [26].

The impact of antimicrobial shortages in EU/EEA (European Economic Area) countries, more specifically in Portugal, has not been reported, but it certainly poses a challenge for doctors, nurses, and pharmacists responsible for guiding on alternative treatments [11]. 

The WHO [23] recommends BPG replacement in cases of a lack of or sensitivity to the drug. In pregnant women, the WHO IST guideline suggests monitored use with erythromycin 50 mg daily for 14 days or ceftriaxone 1 g intramuscularly once daily for 10–14 days or azithromycin 2 g once daily, orally. Doxycycline should not be used in pregnant women. The challenge is that these drugs cannot cross the placental barrier and do not treat the fetus. BPG is considered the only antimicrobial to be effective in preventing maternal-to-fetal transmission.

Although syphilis is easily treatable and preventable, it is paramount to screen pregnant women at prenatal visits to ensure timely diagnosis, preferably at the early stages of infection, and timely treatment, up to 30 days before parturition. That will help reduce the risk of adverse fetal outcomes and facilitate infection control [15].

### 1.1. Sociodemographic Aspects of Brazil and Portugal

Socioeconomic factors strongly influence inequalities in public health throughout life. In addition, other structural factors, such as climate change, and environmental threats, also delay steps towards achieving better health standards. Therefore, socioeconomic divergences between countries are determining factors of variations, trends, and indicators of human health and development [27]. 

In Brazil, a developing country, many people are living in situations of social vulnerability (people deprived of their liberty, users of toxic substances; indigenous people, juveniles, pregnant adolescents, immigrants, and health vulnerabilities) [28]. Bezerra et al. [25] correlated the high incidence rates of SC in regions of high social vulnerability. In the northern region of Brazil, high infant mortality rates were observed, and in the northeast, a high rate of fetal loss after the first trimester of pregnancy. Many Brazilians live in an unfavorable social and economic situation, which is strongly reflected in health care [29]. These data suggest a worrying scenario of social vulnerability in Brazilian maternal and child health [25]. In epidemiological analyses, MS is more prevalent in women living in precarious socioeconomic conditions [30,31].

In a survey conducted in 2015 by the Brazilian Ministry of Health among mothers of children diagnosed with congenital syphilis, 78.4% of these mothers sought prenatal care, and 51.4% of the cases were diagnosed. However, more than half of the mothers (56.5%) received inadequate treatment, and 27.3% did not have access to treatment. In addition, the risk of congenital syphilis increases by five times when maternal partners are infected and do not receive treatment (62.3%) [32,33]. 

In this scenario, the need arises to develop essential strategies to identify vulnerable populations and, thus, offer diagnostic tests and adequate treatment for pregnant women and their sexual partners, aiming at ending the CS transmission cycle [5]. 

Other factors can be considered as the cause of the high rates of syphilis in pregnant women and congenital syphilis in Brazil, such as the quality of prenatal care, late detection, inadequate clinical management, resistance or difficulty of pregnant women and their sexual partners in adhering to treatment, in addition to underreporting of the disease [34,35].

The CS incidence rate is a strong indicator of the quality of prenatal care. For example, there is a very close correlation between the incidence rates of stillbirth due to syphilis and inadequate prenatal care [25]. Quality is generated in prenatal care when it is possible to follow the entire process of this pregnant woman until the birth of her baby. However, it is necessary to integrate data into a unified health system in which epidemiological surveillance, primary care for health, notifications, and prevention and diagnosis actions are carried out to achieve the objective of reducing maternal and child mortality associated with this infection [36]. 

A vital intervention strategy to respond to this public health problem is managing the case from prenatal care to the baby’s birth. Case management favors the active search. Therefore, it makes it possible to find pregnant women in the health system and thus monitor the entire pregnancy, an aspect that favors the qualification of prenatal care in the context of maternal and child care lines. Furthermore, it is a fundamental factor in reducing vertical transmission of syphilis and maternal and infant mortality [5,25]. Likewise, it is essential to investigate the relationship between the reported cases of MS, as these cases do not necessarily represent the actual data on the occurrence of this infection. Access to diagnosis and data from the reported data is designed for the proper epidemiological context of the disease. However, the management of syphilis cases is still challenging for the Ministry of Health due to continental dimensions, marked social inequality, and a vast population [5].

The management of cases of pregnant women must be accompanied by a multidisciplinary health team composed of managers, doctors, social workers, psychologists, nurses, and health agents who work in the community. This multi-professional team is necessary so that the management of cases occurs integrally until the moment of parturition; when aligned with the active search, it will contribute to mitigating the effects of the social vulnerability of these pregnant women, who were eventually infected by Treponema pallidum during pregnancy gestation. Therefore, they will be more likely to undergo adequate treatment throughout the prenatal period, an aspect that will lead to a reduction in congenital syphilis rates, that is, vertical transmission of syphilis [37,38,39].

Another worrying factor is the process of notification and management of syphilis cases in Brazil, which still requires significant improvements and investments. The three forms of syphilis were included in the group of notifiable infections in 1986 (CS), 2007 (MS), and 2010 (acquired syphilis) and are made through the Information System of Notifiable Diseases of the Ministry of Health of Brazil [40,41]. However, the Brazilian notification process is still flawed, as it generates fragmented data and contains unjustifiable delays [42,43,44]. For example, in Brazil, the consolidation of SINAN data regarding syphilis occurs only once each year, and it is always the data from the previous year. This makes decision-making much more complex regarding the conduct of public health policy to respond to the syphilis problem in the country. Furthermore, with a year delay in obtaining consolidated information on syphilis, the public health manager will have to make decisions looking to the past. Therefore, this aggravates public health problems and the general population [33,45].

According to Ordinance 204 of 17 February 2016, syphilis requires mandatory weekly notification, and according to the Unified Health System (SUS) protocols, a flow of data sharing must be followed in the spheres of SUS management. However, this deadline is not obeyed, and this causes a delay at the beginning of treatment; and if the case is a pregnant woman, the situation worsens, as several problems may occur, such as maternal mortality or vertical transmission of syphilis [46,47].

A study by Garbin et al. [47] analyzed 157 compulsory notification forms in which 103 were of acquired syphilis (65.6%), 32 (20.5%) of gestational syphilis, and 12 (14%) of congenital syphilis. However, many forms were incomplete without sociodemographic data. Essential data for acquired syphilis as area of residence (39 reports, 24.8%), level of education (32 reports, 22.1%), patients seen at the health unit (9 notifications, 5.7%), and date of compulsory notification (6 notifications, 3.8%) were lost, in addition to clinical, epidemiological, laboratory, treatment and conclusion data. In this study the authors showed that the predominant age group was 20 to 29 years old, with 33 individuals (32%); 82 individuals (79.6%) were male; 58 were white (65.2%); 44 individuals did not complete high school (57.9%), and 29 lived in urban areas (82.9%). On the other hand, data on gestational syphilis also show the same predominant age group between 20 and 29 years, and 18 individuals did not complete secondary education (66.7%). Regarding the performance of non-treponemal tests, 24 (88.9%) tests were reactive, and only two were non-reactive (7.4%). As for sexual partnerships, 10 (40%) of the partners were not attended. These data are lost over the years, so they directly influence the control of disease cases and the proper conduct of public health policy in response to the problem of syphilis. In addition, the syphilis notification model in Brazil does not follow prenatal care until the baby is born, so notifications of congenital syphilis are fragile, as the health team cannot obtain reliable information about prenatal care. The disease notification system currently used in Brazil is inadequate due to two main reasons: (i) obsolescence of the system and (ii) mainly because it does not consider the management of pregnant women’s cases, a factor that is also determined by the lack of integration between the Surveillance and Primary Health Care [29]. This aspect makes maternal and child care precarious due to the lack of qualified information. 

There is an urgent need for a health information system that (i) integrally manage the case; (ii) links essential areas such as surveillance to primary health care; (iii) monitors the pregnant woman’s prenatal care until parturition; (iv) alerts the manager about the incidence of new cases of the disease; and (v) carry out the notification of the grievance promptly, aiming at improving the quality of care and strengthening decision-making by managers and health professionals [48].

In Portugal, a country with a more stable economy than Brazil, with smaller territorial dimensions and a population of only 5% (approximately 10.800 million inhabitants) [49,50] than the Brazilian population (approximately 214 million inhabitants) [51], it is recommended to perform patient screening in the preconception consultation, and this is repeated in the first prenatal consultation, in the second and third trimesters of pregnancy, obligatorily. Pregnant women who were not previously screened (either because they were an immigrant or because they realized they were pregnant before the preconception consultation) or who had a positive test in the first trimester should be retested after parturition [52].

In the country, according to the General Directorate of Health, the entire population has access to the services provided by the National Health Service [53]. In addition, contraception fees and family planning programs are accessible to the entire population. Infant mortality in Portugal is estimated at 4.4 deaths/1000 live births. However, maternal mortality is not a public health problem in the country and has significantly reduced in recent years. The number of registered health professionals (doctors, nurses, dentists, psychologists) is considered satisfactory, besides having a well-structured STI surveillance and control system. The system of notifications generates alerts, communicated via email or text message over the phone to health authorities to adopt strategic prevention and control measures. In addition, the National Epidemiological Surveillance System (SINAVE) also communicates with laboratory notifications. When completing the notification, if there is no information, automatic alerts are launched to local and regional health authorities and the General Directorate of Health [54,55].

In 2014, the system was implemented in 25 hospital units, with 93% of all AIDS patients being followed up in mainland Portugal. The system also provides a multidisciplinary approach, verifying the need for the intervention of different professionals when it comes to sexually transmitted infections. The system is interconnected with other levels of health care, allowing physicians to enter mandatory reporting of diseases, and the entire clinical process of the patient is monitored at local, regional, and national levels [54].

The difference in contexts between Brazil and Portugal is evident due to the adoption of practical measures that bring good results in controlling and combating syphilis in Portuguese public health, so there is a need for analysis and studies of the adoption of clinical protocols that can help in the management of syphilis cases in Brazil.

### 1.2. Clinical Protocols and Treatment Guidelines (CPTG)

When adopted by healthcare facilities, clinical protocols and treatment guidelines assist healthcare providers and managers in diagnosing, treating, and monitoring pregnant women and their babies—in addition to supporting efforts to prevent new infections. Given the harmful fetal outcomes, mandatory reporting of CS and MS and timely interventions for the pregnant women, are strongly recommended. Moreover, clinical and demographic data—i.e., stage of disease, pregnancy week, syphilis screening results, treatment adopted, drug use, HIV testing, treated or untreated partners—should be collected based on guidelines versed in health protocols that aid in pregnancy and postpartum follow-ups [56,57].

However, currently established guidelines and accurate diagnosis still has some shortfalls to be addressed and then effectively applied, especially concerning tests in which patients’ results have low or inconsistent positive titers, along with difficulties in diagnosis interpretation as seen with asymptomatic infants, thus delaying prompt medical intervention and, consequently, timely treatment [58]. 

Published in 2017, WHO guidelines on syphilis screening and treatment for pregnant women [17] recommends that all pregnant women should be screened for syphilis at their first prenatal visit, preferably in the first trimester, then rescreened in the third trimester, at the time of parturition, and in cases of miscarriage or stillbirth. The protocol also advises that a rapid test be performed in areas with high syphilis prevalence (5% or more). If the result is positive, the first dose of BPG should be administered [57].

The Directorate General of Health Services (DGS)—the body that implements surveillance, information, and health promotion measures and coordinates actions proposed by the Ministry of Health for European and international relations—gives similar recommendations for monitoring pregnant women during prenatal care (PC) visits. In addition, high-income countries belonging to the European Union (EU) follow a PC program that includes at least one or two syphilis screening tests throughout pregnancy [11,59,60,61].

This paper compares Brazilian and Portuguese guidelines for monitoring and treating MS and CS to assess syphilis clinical management in both countries and investigate potential gaps in protocols applicability towards adequate adherence by health care systems. In addition, we evaluated data on MS and CS new infections in Brazil and Portugal.

## 2. Materials and Methods

### Study Design and Search Strategy

This review was carried out by bibliographic research in two public databases (Google Academic and PubMed) and government websites from both countries, where CPTG was selected for CS and MS, published between 2007 and 2021. It is a narrative review of descriptive character through data analysis. Protocols that did not contain the terms “syphilis, “maternal syphilis,” and “congenital syphilis” were excluded. Data from reports from health agencies in Brazil (Ministry of Health) and Portugal (General Directorate of Health) were also analyzed [62,63]. 

## 3. Results and Discussion

### 3.1. Current Epidemiology

In Brazil, the average detection rate of MS in 2020 was 21.6 cases per 1000 live births (against 20.8 cases in 2019); and for CS, 7.7 per 1000 live births (against 8.5 in 2019) [4,64]. These results were observed after the “Syphilis No! Project” (SNP) developed a variety of actions to fight the disease. In 2018, the Brazilian Ministry of Health (MoH) partnered with the Pan American Health Organization (PAHO) and the Federal University of Rio Grande do Norte (UFRN) through the Laboratory for Technological Innovation in Health (LAIS) to implement the SNP nationally [2,5,31,65].

The project’s goal is to strengthen health care networks to fight syphilis by bolstering access to diagnosis and adequate and timely treatment, as well as adopting strategies toward educommunication, social interventions, and public health campaigns targeted at the general population and health workers and managers [2,57,59].

Like Brazil, EU nations have effective health programs to eliminate congenital syphilis. For example, in Portugal, a relatively steady incidence rate of CS has been observed over the years, with approximately three confirmed cases per 100,000 population in 2018. Hence, numbers have been consistently low and are in line with the other EU and EEA (European Economic Area) countries [66,67].

Despite the low CS incidence in Portugal, some cautions regarding pregnant women’s social, economic, and behavioral factors are directly related to case incidence. Examples include sexual behavior, drug use, living in custodial settings, low income, low literacy, lack of syphilis screening during pregnancy, absent or inadequate treatment, and infection after a preliminary negative test [11]. It is worth noting that CS is a notifiable disease in both countries we analyzed; hence physicians and other health practitioners, or those responsible for public or private health services, must notify health authorities of any case of syphilis [11,68].

### 3.2. “Syphilis No!” Project (SNP)

The project aims to ensure access to care in health networks underpinned by robust strategies involving campaigns, communication, massive health education, and health interventions that pave the way for prompt care, diagnosis, and close monitoring of patients with syphilis [2,57,59,65]. As a result, the project has positively transformed the landscape of syphilis in Brazil, where a drop in the number of new infections has been observed in recent years, especially when it comes to CS, which had persistently high rates for more than a decade [64].

Although the SNP has a national reach, its effects, unfortunately, are uneven across Brazilian regions. That is because each health region’s socioeconomic and cultural peculiarities, not counting the continental size of Brazil, directly influence the response to infectious diseases [31].

Based on that, demographic information and analysis are indispensable for decision-making, as territorial dimension, low literacy and education levels, the number of children per woman, and unemployment rates are strictly related to the incidence and management of syphilis cases in the countries analyzed [69], (see Figure 1).

### 3.3. Guidelines

#### 3.3.1. Portuguese Guidelines

In Portugal, current strategic approaches to maternal and newborn care followed in primary health care (PHC) to fight syphilis are regulated by EU health agencies (European Centre for Disease Prevention and Control–ECDC) and DGS. Annual reports issued by ECDC provide information on the current status of CS, the increase or decrease in the toll of cases, and the measures to be undertaken to control the disease. However, these documents may underreport the condition. For instance, some Member States fail to notify cases, raising concerns for health regulators, as low rates could be due to reduced reporting by primary health centers rather than an actual drop in syphilis cases [67].

In Portugal, the periods for HIV, syphilis, and rubella serology testing during preconception and conception are regulated. These indicators are registered in the National Health Service platforms and then audited. At PHC, the system contemplates routine screening tests to be requested for people who are pregnant or might become pregnant [70].

Portuguese protocols stipulate that syphilis screening is compulsory during pregnancy’s first and third trimesters. Routine testing uses the nontreponemal tests—venereal disease research laboratory (VDRL) or the rapid plasma reagin (RPR)—and a treponemal test—treponema pallidum particle agglutination (TP-PA) assay, Fluorescent Treponemal Antibody Absorbance (FTA-ABS), or a chemiluminescence immunoassay (CLIA). Nontreponemal tests are used for screening and treatment follow-up, while treponemal tests are confirmatory (Figure 2) [70,71].

#### 3.3.2. Brazilian Guidelines

The latest Clinical Protocol and Treatment Guidelines for preventing mother-to-child transmission (MTCT) of HIV, syphilis, and viral hepatitis, released in 2019, aims to enhance and operationalize the management of these infections, including CS and MS, which are prioritized in this protocol. Hence, the document guides managers and health workers on specific behaviors around diagnosis, treatment, and control of the aforementioned STIs in target populations. In addition, the previous year’s CPTG (2018) featured a chapter focused on sexual health for people with active sex life. It was meant to support healthcare professionals’ practice and align it with current tools for prevention guidance [72].

According to those protocols, the approach proposed by the Brazilian MoH is to screen pregnant women for syphilis at the first prenatal visit, preferably in the first trimester of pregnancy, and to rescreen them in the 28th week and at the time of parturition (Figure 2). Syphilis screening is also recommended in cases of diagnosis of both miscarriage and stillbirth [73]. Equally important, the MoH recommends protocols and diagnostic routines for babies born to women diagnosed with syphilis in pregnancy. Factors considered are early diagnosis and treatment adequacy for syphilis; clinical, laboratory, and radiographic evidence of syphilis in the newborn; and comparison of the mother’s VDRL titers with the baby’s test at birth [74].

The MoH, through the Department of Sexually Transmitted Infections, AIDS (Acquired immunodeficiency syndrome) and Viral Hepatitis, recommends prompt treatment and exclusive use of rapid tests under the following conditions: (i) health service provision sites with insufficient infrastructure or in areas of difficult access; (ii) people at high risk of infection by *T. pallidum*; (iii) riverine and indigenous peoples; (iv) people whose partners have been diagnosed with syphilis; (v) people living with HIV/AIDS; (vi) sexual violence victims; (vii) people diagnosed with viral hepatitis; (viii) pregnant women during hospitalization for parturition in maternity hospitals; and (ix) miscarriage, regardless of gestational age.

Some factors, such as the absence of effective prevention and control measures, socioeconomic, demographic, and welfare aspects, continue to spur syphilis cases persistence in the country, even though the infection is easily detectable and has a relatively low-cost treatment available through the Brazilian National Health System (SUS). Finally, delayed diagnosis and absent or ineffective treatment for pregnant women contribute to the escalating incidence of CS [75].

### 3.4. Protocol Gaps

MS has become a significant public health challenge as several countries face a high incidence of CS. Even though the infection is predictable and preventable, the treatment has a low cost, and public health agencies provide screening tests [17,75,76,77,78].

In a 2010 initiative to eliminate CS, PAHO established the goal to curb the incidence rate of CS to a rate equal to or lower than 0.5 cases per 1000 live births by 2015. However, in Brazil, numbers are still striking, probably due to socioeconomic factors and inequalities directly affecting case incidence. According to recent research, the root cause of the high incidence of CS worldwide, alongside other factors, is intrinsically related to inadequate PC [78,79]. However, some sociodemographic indicators may also contribute to the current syphilis scenario in both countries, such as level of education, mortality/birth rate, poverty- and age-related factors, and drug use [80,81,82]. Previous research has shown that the quality of care provided to pregnant women during prenatal and childbirth care reduces MTCT. Syphilis control is intricately related to serological screening and adequate treatment of mothers and partners.

Portugal has solid family planning programs in which sexually active women are submitted to several routine screening tests, including syphilis. Through family planning, it is possible to establish an early diagnosis of syphilis, facilitating treatment and eliminating syphilis or any other infection during the preconception period [83]. That is nonetheless the standard approach followed at primary health centers. However, in some instances, patients may arrive at a health center already pregnant and infected or be immigrants who have not received adequate PC in their country of origin or use drugs. In such situations, the protocol is to treat and monitor pregnant women until parturition; the baby when it is born, and the partner if they authorize it [70].

Despite Brazil’s comprehensive PC coverage, this stage often lacks uniformity or adequate functioning since CS stems from undiagnosed or ineffectively treated maternal syphilis [73]. A lack of PC [59,60,61], lack of treatment, or inappropriate treatment during pregnancy [84]—and occasionally syphilis infection that went undetected in prenatal visits—are determining factors for the incidence of CS cases in Brazil.

In recent years, several versions of Brazilian and European (Portuguese) clinical protocols have been released, evincing dilemmas and uncertainties that new approaches have posed for syphilis screening and the need to incorporate new practices that adapt to each region’s reality. Unfortunately, these gaps result in high costs, unnecessary hospitalizations and inappropriate treatments, family disruption, and misdiagnoses. Currently, two approaches are being used to screen for syphilis in pregnant women: the traditional algorithm and the reverse algorithm. Screening by the traditional algorithm starts with a serological non-treponemal test (NTT); if this is reactive, a treponemal serological test (TT) will be performed to confirm positivity. In the reverse algorithm, the first serological test performed is a TT, and if this is reactive, positivity must be confirmed by a quantitative NTT. When TT is reactive, and NTT is nonreactive, a different TT is performed to resolve the disagreement [85]. Both algorithms have advantages and limitations, but both are reliable. The traditional algorithm presents low cost and high reliability in low incidence scenarios of syphilis, while the reverse algorithm. However, it has a higher cost, as it requires a more significant number of confirmatory tests, has greater sensitivity, and improves the chances of detecting latent and early primary syphilis [85,86,87].

Given this scenario, it can be said that although Brazil and Portugal use different approaches in screening for syphilis in pregnant women (Figure 2), this does not impact the accuracy or precision of the diagnosis.

Notwithstanding high numbers of CS and MS cases, Brazil has robustly invested in health promotion campaigns to combat syphilis since 2018, including educational activities targeting disease control [88]. New strategies should focus on comprehensive efforts toward infection prevention and control to strengthen new approaches and facilitate access to information, treatment, and long-term care to ensure that adherence to long-term follow-up. That could also mitigate the widespread stigma and discrimination around syphilis.

In Portugal, sexual education, family planning, and reproductive health are increasing and continuous; these factors have also collaborated with the control of sexually transmitted infections [89].

The difference in the incidence of cases between the selected countries brought the possibility of evaluating the conduct being adopted in this place, which may reflect on the control of the disease. Although we know the social and economic vulnerabilities in which many Brazilian families are inserted and do not have the necessary resources to face risks and diseases, it is known that it is possible to spread knowledge about the disease through public health policies, campaigns encouraging testing, treatment and the practice of healthy habits. It is believed that to control and eradicate diseases, measures that are interconnected in the social, economic, and public health spheres are necessary [89,90,91]. Therefore, professionals and health units must adopt specific prevention actions aimed at each case’s economic and social reality, aiming at strategies for the control and eradication of syphilis in Brazil and the world.

The data obtained in this work can support health professionals and managers in conducting and managing syphilis cases in pregnant women.

## 4. Conclusions

This study analyzed strategies to control and combat CS and MS in distinct countries (Brazil and Portugal). Despite the contrasting statistics, both countries adequately conduct case management of syphilis in pregnant women, their partners, and babies through clinical protocols standardized by regulatory agencies and used in health centers according to each country’s reality. Nevertheless, MS and CS incidence in Brazil is substantially higher than in Portugal, suggesting that this discrepancy lies in differences in economic, cultural, and social status, and notably territorial aspects (i.e., the continental size of Brazil), rather than differences in approaches and efforts for disease control. Factors that negatively influence efforts to tackle the disease as they directly affect the quality of family planning and PC, which is determine for cases to drop, both for CS and MS cases. These findings can collaborate with the control of the syphilis epidemic in Brazil to correlate data with countries where the disease scenario is different. Through these behaviors, we direct correct decisions for the monitoring and treating the disease in pregnant women and children with congenital and, in this way, we achieve a significant reduction in the incidence of syphilis cases in Brazil.

With this, it is suggested that future studies can, through these findings, collaborate with the control of the syphilis epidemic in Brazil by correlating data with countries where the disease scenario is different. Furthermore, these behaviors can lead to correct decisions for monitoring and treating the disease in pregnant women and children with congenital diseases. In this way, we will significantly reduce the incidence of syphilis cases in Brazil. In addition, the conducting research to assess other socioeconomic indicators that may influence the number of syphilis cases in the country. Thus, these results become a relevant initiative for adopting public policies and strategies to control and combat the infection in these countries.

## Figures and Tables

**Figure 1 ijerph-19-10513-f001:**
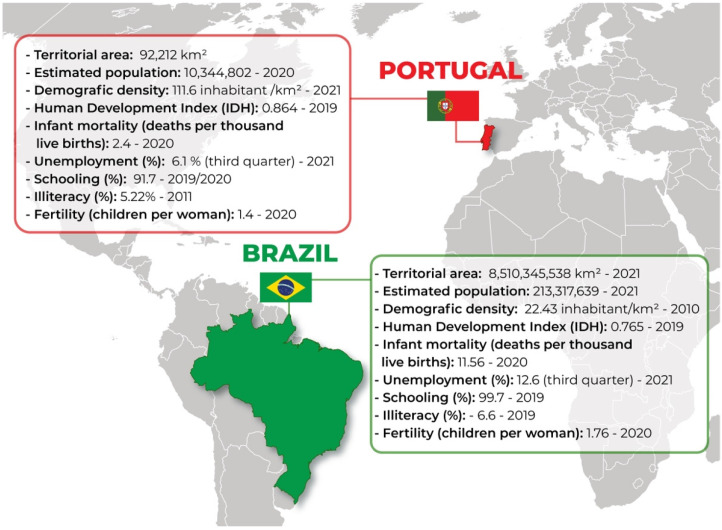
Sociodemographic data from Brazil and Portugal. The map shows socio-economic indicators for each country—Source: authors’ elaboration.

**Figure 2 ijerph-19-10513-f002:**
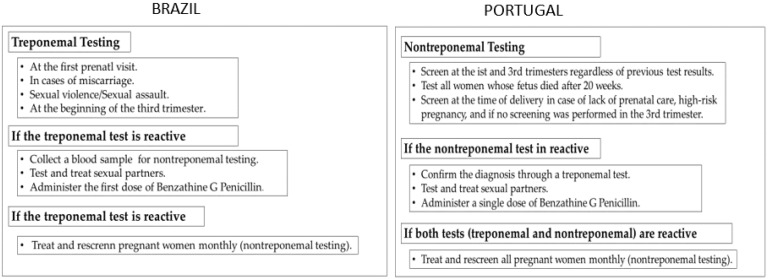
Flowchart of procedures for diagnosis and treatment of MS in Brazil and Portugal. The figure demonstrates the testing and treatment guidelines adopted in each country. Source: authors’ elaboration (Brazil: DCCI/SVS/MS; Portugal: DGS).

## Data Availability

Data sharing not applicable.

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
