# Peer review of "Clinical Protocols and Treatment Guidelines for the Management of Maternal and Congenital Syphilis in Brazil and Portugal: Analysis and Comparisons: A Narrative Review"

_ijerph, 2022, doi:10.3390/ijerph191710513_

Round 1
Reviewer 1 Report
Dear authors,
I have placed my comments in the attached document.
I wish you the greatest success for your article.
Conceição Santiago

Author Response
Dear Reviewer 1,
Please see the attachment.
Kind regards,
Talita Brito

Reviewer 2 Report
This paper deserves to be published, the topic is relevant and gives a contribution to strengthen quality within primary health care services, having said so there are some major revisions needed. The sociodemographic data from Brazil and Portugal needs to be more elaborated in the introduction as an overall theoretical framework. Further a broader framework is needed that interconnect sociodemographic data, maternal and congenital syphilis and how these guidelines secure access to health care services for all citizens and with a specific focus on vulnerable women. The paper needs to address how to reach women who do not necessarily come to health care facilities and are in risk off or are being infected before or during pregnancy. The authors could discuss and include a broader review in how the guidelines incorporate management of clinics that secure quality of access to health services – and interconnect the guidelines review to universal health coverage (WHO). This means among other issues to discuss anonymity, opening hours, location, financial payment, friendly and ethical sound environment in the management of clinics by health care professional. The paper would be even more relevant if it include an broader environmental health approach on how this guideline address these matters in the two countries. If the guidelines do not include these matters - the authors could discuss these matters more in depth. Include these matters in the comparison of the two countries when it comes to STD and sexual health clinics for youths, women’s, adults (both men and women) etc.
I recommend a broader discussion and review processes around access for all and how to include those women and males that are in risk.
Author Response
Dear Reviewer 2,
Please see the attachment.
Kind regards,
Talita Brito

Round 2
Reviewer 2 Report
The revised article seemed to be sound according to the objective and purpose.
I recommend the article to be published.